# Strategies to Improve the Transdermal Delivery of Poorly Water-Soluble Non-Steroidal Anti-Inflammatory Drugs

**DOI:** 10.3390/pharmaceutics16050675

**Published:** 2024-05-16

**Authors:** Alexandra Balmanno, James R. Falconer, Halley G. Ravuri, Paul C. Mills

**Affiliations:** 1School of Veterinary Science, The University of Queensland, Gatton Campus, Gatton, QLD 4343, Australia; alexandra.lonergan@uq.net.au; 2School of Pharmacy, The University of Queensland, Dutton Park Campus, Woolloongabba, QLD 4102, Australia; j.falconer@uq.edu.au; 3School of Biomedical Sciences, The University of Queensland, St Lucia, QLD 4072, Australia; h.ravuri@uq.edu.au

**Keywords:** transdermal, microemulsion, NSAID, nanoformulation

## Abstract

The transdermal delivery of non-steroidal anti-inflammatory drugs (NSAIDs) has the potential to overcome some of the major disadvantages relating to oral NSAID usage, such as gastrointestinal adverse events and compliance. However, the poor solubility of many of the newer NSAIDs creates challenges in incorporating the drugs into formulations suitable for application to skin and may limit transdermal permeation, particularly if the goal is therapeutic systemic drug concentrations. This review is an overview of the various strategies used to increase the solubility of poorly soluble NSAIDs and enhance their permeation through skin, such as the modification of the vehicle, the modification of or bypassing the barrier function of the skin, and using advanced nano-sized formulations. Furthermore, the simple yet highly versatile microemulsion system has been found to be a cost-effective and highly successful technology to deliver poorly water-soluble NSAIDs.

## 1. Introduction

Non-steroidal anti-inflammatory drugs (NSAIDs) are commonly used to manage pain and inflammation in humans and animal species, and are particularly useful for chronic inflammatory conditions, such as osteoarthritis. However, their oral administration often leads to adverse effects, particularly related to the gastrointestinal (GI) system [1]. NSAIDs that target the COX-2 isoenzyme, termed the coxibs, have fewer adverse effects compared to traditional non-selective NSAIDs, but may still cause GI-related adverse events when taken orally due to high local levels of the drug in the stomach [2,3]. The transdermal delivery (TD) of NSAIDs has been used to avoid these effects, and offers many advantages over oral administration, including being non-invasive, leading to fewer GI adverse effects, and avoiding hepatic first-pass metabolism [4,5]. The principles of TD have been reviewed extensively elsewhere and will not be covered in this review [5,6]. Similarly, recent reviews have described strategies to enhance the penetration of NSAIDs for local effects [5,6], but there is limited discussion of approaches to achieve therapeutic systemic concentrations of NSAIDs.

The successful delivery of drugs through the skin is influenced by multiple factors relating to the skin, the drug, and the delivery system. One of the main challenges with the transdermal delivery of many NSAIDs, but particularly the COX-2-selective class, is poor water solubility, leading to difficulty incorporating sufficient quantities of the drug into the delivery system or vehicle, which decreases the bioavailability of the transdermal drug and may require a higher dose to be applied to achieve the desired therapeutic effect. This challenge is increased if systemic therapeutic concentrations are desired. In addition, poor permeability of the drug through the skin can also lower transdermal bioavailability. To improve bioavailability, various strategies can be utilised to increase the solubility of the drug in the formulation and the permeation of the drug across the skin, including modifying or bypassing the stratum corneum (SC), modifying the drug molecule, and modifying the vehicle. Recent advances in nanotechnology have led to increased interest in the application of nano-sized carrier systems, such as liposomes, niosomes, transfersomes, microemulsions (MEs), and nanoparticles, in TD systems [7,8,9]. This review will highlight the various strategies employed with poorly water-soluble NSAIDs to improve their solubility in TD systems and their permeation through the skin.

## 2. Enhancement of Transdermal Drug Delivery

### 2.1. Chemical Penetration Enhancers

A chemical penetration enhancer (CPE) is a compound that helps to increase the partitioning of the drug into the skin through an interaction with the structure of the SC. Enhancing skin permeation using CPEs is one of the most commonly employed methods in formulation development, with nearly every topical formulation on the market containing at least one penetration-enhancing agent. The ideal CPE has the following properties [10]: It is pharmacologically inert and chemically stable;It is physically and chemically compatible with all other components of the formulation (excipients and drug/s);It is non-irritating, non-allergenic, and non-toxic;It has a rapid onset with a predictable and suitable duration of activity;It has completely reversible effects upon removal, with no permanent damage to the protective barrier of the SC;It is cosmetically acceptable when applied to the skin;It is odourless, inexpensive, tasteless, and colourless.

CPEs increase drug permeation by interacting with the multilaminar lipid domain of the SC or interacting with the intercellular proteins. In the lipid domain, there are three main sites with which a CPE may interact: (1) the polar head groups of the lipids; (2) the aqueous domain within the lipid layers, and (3) the lipid alkyl chains [11] (Figure 1).

Interaction directly with the polar head groups, generally through hydrogen bonding, causes disorder and disruption within the structure, allowing excessive fluidisation of the lipid regions. The increased water between the layers typically results in decreased diffusion resistance and increased flux for both hydrophilic and lipophilic solutes [12]. The CPEs laurocapram (Azone^®^) and isopropyl myristate (IPM) both utilise this mechanism of action, although IPM also interacts with the lipid alkyl chains [13,14,15,16].

The hydrophilic, aqueous region of the lipid domain is the primary site of action for some CPEs, such as propylene glycol (PG), diethylene glycol monoethyl ether (DGME), and ethanol. These chemicals increase penetration by accumulating in the hydrophilic region which changes the solubility of the drug in this layer [4,12,15].

The final site of action is in the lipid tails of the hydrophobic (lipophilic) region. By inserting themselves between the lipid tails, these CPEs are able to disrupt the normal orderly arrangement of the lipids, which promotes the fluidisation of this area, and thus alters drug penetration [12]. Saturated long-chain fatty acids of 9–14 carbons and unsaturated long-chain fatty acids of 18 carbons (such as oleic acid (OA)) have been found to be optimal for use as CPEs acting at this site [12,17].

**Figure 1 pharmaceutics-16-00675-f001:**
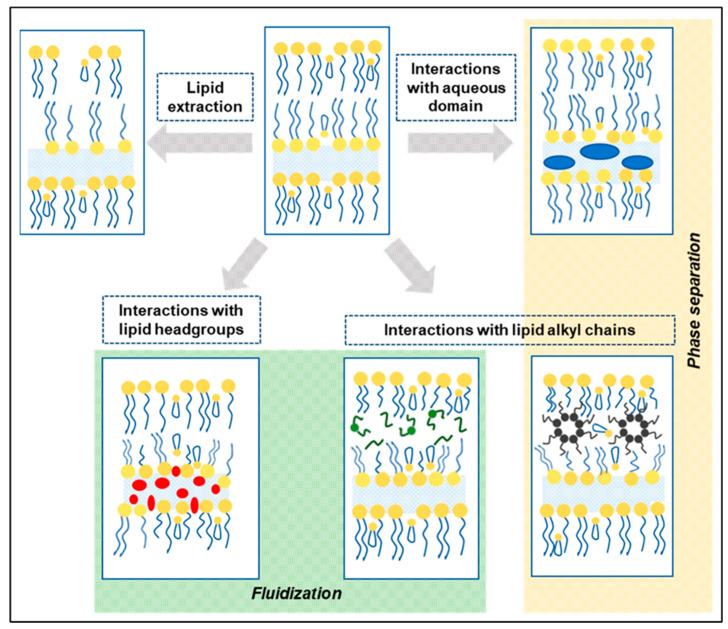
Representation of possible interactions that chemical permeation enhancers may have on the multilaminar lipid domain of the SC. Image from Pereira, Silva, Pinheiro, Reis, and Vale [18] used with permission under Creative Commons licence (https://creativecommons.org/licenses/by/4.0/).

#### 2.1.1. Categories of Chemical Penetration Enhancers

##### Alcohols

Alcohols are commonly used in transdermal formulations for their penetration-enhancing ability, with short-chain alcohols, such as ethanol and isopropanol, being popular choices [16].

Ethanol and isopropanol are short-chain alcohols that are commonly used as solvents or co-solvents in transdermal formulations. These alcohols elicit their permeation-enhancing action through various mechanisms. In the vehicle, they can act as a solvent to increase the solubility of the drug in the donor phase. In addition, being a volatile solvent, the thermodynamics of the drug in the formulation can be increased as the alcohol evaporates [16,19]. In the SC, alcohols can extract lipids, resulting in increased water in the lipophilic region between layers [20]. It has also been suggested that ethanol and isopropanol can accumulate within the hydrophilic domain, which may increase the solubility of drugs in this area [16,21]. Ethanol has been used to increase the penetration of NSAIDs, such as piroxicam [22] and meloxicam [23], as well as many other lipophilic compounds.

PG is a glycol alcohol that acts by inserting itself and accumulating in the hydrophilic regions of lipid layers, causing disorder [24]. The insertion of PG also alters drug solubility in the skin so that it matches that of the drug in the vehicle, leading to increased partitioning of the drug from the vehicle into the SC [11,15,16,21,25]. PG is used in a wide number of transdermal drug permeation studies, either alone or in combination with other excipients and CPEs. Synergism has been displayed when it is used in combination with other CPEs, especially highly lipophilic enhancers such as OA, Azone^®^, isopropyl myristate, and terpenes [12,26,27,28,29]. PG has been effectively used to enhance the penetration of highly lipophilic drugs [27] and NSAIDs, such as meloxicam, lornoxicam, and celecoxib [23,30,31].

DGME, also known as Transcutol^®^, is an ether alcohol that acts by partitioning into the aqueous domain of intercellular lipids. Osborne and Musakhanian [32] reported that the application of Transcutol^®^ to the skin helps to maintain a high level of hydration, which improves skin permeability and drug solubility in the SC, rather than causing disruption and fluidisation of the structured lipids (unlike Azone^®^ or OA). Transcutol^®^ has been shown to act synergistically with Azone^®^ [32]. It has been successfully used to increase the permeation of a variety of NSAIDs, such as celecoxib [30,33], ibuprofen [33,34], and meloxicam [23,35].

##### Fatty Acids and Fatty Acid Esters

Fatty acids have been used to increase the permeation of a wide variety of drugs. Saturated long-chain fatty acids of 9–14 carbons and unsaturated long-chain fatty acids of 18 carbons (OA) have been found to be the most successful penetration enhancers [12,17].

OA is an unsaturated long-chain (C18) fatty acid which exerts its effect by inserting itself between alkyl lipid tails, disrupting the lipid packing and causing fluidisation of this region [16,36]. In addition, OA may accumulate in pools within the lipid layer, causing further disruption to the normal structure [16,37]. OA is an effective CPE for a variety of drugs across the skin, including many NSAIDs (meloxicam [35], tenoxicam [38], piroxicam [39], celecoxib [40], lumiracoxib [41]). The combination of OA, polysorbate (Tween^®^ 80), and isopropanol was found to enhance the ex vivo permeation of risperidone through human skin [42]. Other fatty acids, such as lauric acid, have been demonstrated to be highly efficacious in enhancing the permeation of highly lipophilic drugs [27].

Isopropyl myristate (IPM) is a fatty acid ester that inserts itself into the lipid domain, which increases the density of lipid packing and decreases the diffusion coefficient and may also interact with polar head groups by attaching to them [14,15]. Pre-treatment of the skin with IPM alone did not provide any permeation enhancement of hydrocortisone, but was synergistic when the solvent isopropyl alcohol (IPA) was added [14,15]. Synergy with N-methyl pyrrolidone has also been suggested [43]. Formulations containing 45% IPM, along with ethanol and eucalyptus oil, were found to enhance the transdermal permeation of ketoprofen through cattle skin [44]. The flux of lornoxicam through rat skin was enhanced when using 20% IPM in combination with Eudragit^®^ E100 and ethyl cellulose [45], and concentrations of 1–4% IPM were able to enhance meloxicam permeability in transdermal patches containing cellulose gelling agents, ethanol, methanol, and glycerine [46].

##### Sulfoxides

The most frequently used sulfoxide compound is dimethyl sulfoxide (DMSO), which acts via an interaction with SC lipids and by denaturing keratin proteins in the SC [47,48]. DMSO has been used extensively to enhance the permeation of many classes of drugs and is included in numerous FDA-approved products, including a diclofenac sodium topical solution (Pennsaid^®^) [49]. In addition, the transdermal permeation of celecoxib through pig skin was enhanced using 5% and 10% DMSO formulations [50]. However, when used at high concentrations, DMSO has the potential to cause skin irritation and potential damage, leading to other sulfoxides, such as decyl methyl sulfoxide (DCMS), being developed [47]

##### Laurocapram (Azone^®^)

Laurocapram (Azone^®^), a chemical hybrid of an amide and sulfoxide, was the first agent specifically designed for transdermal penetration enhancement [51]. Laurocapram is highly lipophilic and is soluble in alcohols, PG, and most organic solvents, and can promote the flux of both lipophilic and hydrophilic drugs [12]. Despite being extensively studied, its mechanism of action is not fully understood, but it has been suggested that laurocapram inserts itself into the lipid domain and then interacts with skin ceramides through hydrogen bonding, resulting in disruption and subsequent fluidisation of the lipophilic domain [16,19,51,52]. The presence of laurocapram-forming pools within lipid regions has also been demonstrated, which may enhance drug partitioning [53]. The greatest enhancement effects are achieved when laurocapram is used in low concentrations (0.1–5%) [19] and will enhance the penetration of NSAIDs, such as naproxen [54], and synergises with other CPEs, such as PG [55], DGME [54], and ethanol [56].

##### Surfactants

Surfactants are a group of chemicals characterised by a polar head with a non-polar tail. Due to the structure of the surfactant, the molecules will gather at the surface of high-surface-tension solvents and can form micelles when at sufficient concentrations [57]. Surfactants can be further classified into four groups based on the nature of their polar head group: non-ionic, cationic, anionic, and amphoteric (zwitterionic). The mode of action varies depending on the surfactant and its classification, with most surfactants generally interacting with both the proteins (especially keratin in corneocytes) and the lipid domain [16].

Non-ionic surfactants are commonly used in many pharmaceutical formulations, as they are generally considered to be safe with lower potential for irritation [58]. Polysorbate 80 is one example of a commonly used non-ionic surfactant, although many products are available. It elicits its action through two proposed mechanisms: by penetrating into the intercellular lipids to fluidise the lipid domain, and by binding to keratin protein, leading to disruption of the corneocytes [16,59]. The structure of polysorbate 80, which has both lipophilic and hydrophilic characteristics, allows this CPE to partition easily between the domains of the lipid layers [59]. Cappel and Kreuter [60] reported a synergistic effect of using polysorbate and PG together, and Shahi and Zatz [61] found that using an isopropanol–water mix with polysorbate increased the flux of hydrocortisone through mouse skin. Polysorbate 80 (Tween^®^ 80) was able to enhance the permeation of meloxicam through rat skin [35], and when used in combination with other CPEs, has been demonstrated to increase celecoxib solubility in microemulsion gel (MEG) formulations [30].

Cationic surfactants, such as benzalkonium chloride (BKC), and anionic surfactants, such as sodium lauryl sulphate (SLS), are also used in transdermal formulations, although they may be potentially more irritating to the skin compared to non-ionic surfactants [58]. In addition, ionic surfactants may interact with ionizable functional groups of the active drug, which may interfere with ME formation [62].

##### Terpenes

Terpenes are a group of compounds derived from essential oils and medicinal plants [63]. In general, terpenes are considered safer than synthetic CPEs, and many are listed on the GRAS (Generally Regarded as Safe) list [47]. The main mechanism of action for terpenes is their interaction with or extraction of intercellular lipids, or their disruption of the packing within the lipid layers, which causes an increase in drug diffusion [26,63]. In a study by Narishetty and Panchagnula [64], it was suggested that terpenes cause SC barrier disruption through three main mechanisms: disordering the lipid chains, disrupting interlamellar hydrogen bonding at the polar head groups, and increasing hydration within the lipid domain. The authors also suggested that hydrocarbon terpenes (limonene) mainly act on lipid alkyl chains, whereas oxygen-containing terpenes (1,8-cineole, menthol) act through all three mechanisms. PG has been shown to synergistically increase the permeation effect of monoterpene CPEs, like 1,8-cineole [26]. The transdermal permeation of naproxen was significantly enhanced using the terpene eucalyptol [65]. Ketoprofen delivery was enhanced in both cattle [44] and dogs [66]. Additionally, eucalyptus oil has been used in MEG formulations to enhance the transdermal permeation of meloxicam through rat skin [67]. Other examples of terpenes used to enhance transdermal NSAID permeation are menthol [68], limonene [69], and thymol [70].

##### Other CPEs

This review covers many of the common classes of CPEs; however, this list is not exhaustive, and there are many other classes of CPEs are not covered, including, but not limited to, pyrrolidones, phospholipids, amines, amides, and other esters.

##### Synergism

Synergistic effects can be achieved when using certain combinations of penetration enhancers. There are many techniques used to exploit the synergistic behaviour of various penetration enhancers, such as co-solvency; creating eutectic mixtures; using vesicles such as liposomes, transfersomes and niosomes; and creating MEs [71,72]. PG, one of the most commonly used penetration-enhancing agents, is synergistic when combined with terpenes, OA, Azone^®^, and IPM [12,63]. Ethanol has demonstrated synergistic behaviour when used with linoleic acid for the TD of lidocaine [73]. A combination of 5% OA and 5% DGME (Transcutol^®^ P) was able to synergistically enhance the transdermal flux of oxcarbazepine through human skin compared to OA or DGME alone [72]. Synergism has also been demonstrated when CPEs are used in combination with physical penetration-enhancing methods [74], such as iontophoresis [63,75,76,77], sonophoresis [78], and electroporation [79].

### 2.2. Physical Penetration Enhancers

#### 2.2.1. Iontophoresis

Iontophoresis involves the application of an electric current between two electrodes that are applied to the skin, which drives hydrophilic and lipophilic charged (ionic) drugs through the skin, generally through the path of least resistance (e.g., through skin appendages) [80,81,82]. The enhancement effect depends on factors such as the current applied, the drug concentration, the molecular size, and the pH of the applied solution [81,83]. The electrical current used is generally small (0.5 mA/cm^2^) [84], although there is potential for it to cause damage to growing hair as it passes through the follicles [85]. Iontophoresis is considered an expensive approach to penetration enhancement but has been successfully used to increase the penetration of a variety of drugs, including ketamine [86] and NSAIDs such as piroxicam [87], indomethacin, aspirin, and ibuprofen [88].

#### 2.2.2. Sonophoresis

Sonophoresis uses low-frequency energy waves to increase transdermal drug penetration by causing acoustic cavitation in the SC [89,90,91]. Wavelengths of 20 kHz–16 MHz have been used for sonophoresis, although lower frequencies (20 kHz–100 kHz) are the most successful for increasing the transdermal penetration of drugs [91]. The TD of many drugs and macromolecules has been successfully enhanced using this technique, including NSAIDs such as diclofenac [92,93] and celecoxib [94]. Although sonophoresis does reduce the skin’s barrier function, this effect is only transient, because as the water permeability of the skin increases 100-fold following sonophoresis, this quickly reverses after removal of the ultrasound [95].

#### 2.2.3. Electroporation

Electroporation is the application of short (microsecond to millisecond) high-voltage pulses through the skin to create transient microscopic pores in the SC [96]. The force from the electrical field is thought to cause water from both sides of the lipid membrane to meet, forming the pores [96,97]. The formation of these pores allows the passage of macromolecules (for example, vaccines [98] and insulin [99]) and large drugs to pass through the SC that otherwise would not be able to passively diffuse through [96,100]. The creation of aqueous pores also allows smaller, water-soluble molecules to easily traverse the SC layer [96]. Similar to iontophoresis, the electrical field produced in electroporation has the potential to cause thermal injuries to the skin [101]. Electroporation has been used to enhance the anti-inflammatory effect and plasma concentrations of diclofenac when applied topically over the stifle joint in rats [102], and has been shown to enhance the transdermal permeation of meloxicam when used alone or in combination with iontophoresis [103].

#### 2.2.4. Laser Ablation

The application of low-energy lasers to skin can selectively damage the SC, decreasing the barrier function and thus increasing the transdermal permeability of many drugs [104]. The disruption to the SC layer caused by the laser is reversible, and the depth of ablation can be controlled by adjusting the energy of the laser [105]. The laser can be used in full- or fractional-beam modalities, with the fractional-beam mode having several advantages, such as faster healing, as it only damages small areas at the target depth, rather than the entire treated area [105,106]. Lasers act to enhance the transdermal permeation of drugs through three primary mechanisms: (i) direct ablation, where drug passage is facilitated through the creation of microchannels in the skin; (ii) photothermal ablation, where the water and other components in the skin are heated through the absorption of the laser energy, resulting in small burns and disruption of the microstructure of the skin; and (iii) through the creation of photomechanical waves, where mechanical waves are created through exposure of a material in contact with the skin, creating transient pores within the intercellular lipid layer of the SC [105,106]. There are a number of different types of lasers used to facilitate TD, with some common types being the carbon dioxide (CO_2_) laser and the erbium-doped yttrium aluminium garnet (Er:YAG) laser [107]. Low-energy laser ablation has been used to enhance the TD of diclofenac through rat skin in vivo [108]. Similarly, the permeation of diclofenac from a gel was enhanced through ex vivo human skin using laser-induced resonant amplitude waves, a technique based on a photomechanical mechanism [109]. However, the need for specialised equipment limits the ability of this technology to be used outside of large clinics and research centres.

### 2.3. Bypassing the SC

Microneedle arrays are transdermal delivery systems composed of rows of short (50–900 µm) micro-diameter (<300 µm) needles that are applied to the skin surface and will pierce through the epidermis, effectively bypassing the barrier function of the SC [110]. Because microneedles are designed to not reach the dermis layer containing the nerve endings, the application is painless and easy. Microneedles can be solid or hollow and may be coated, uncoated, or dissolving [110]. Because this TD system does not rely on passive permeation through the skin, it is a viable option for larger drugs and drugs that would otherwise be unsuitable for TD due to their physicochemical characteristics. Arrays of chitosan microneedles have been demonstrated to deliver meloxicam into the ear skin of cattle [111]. Dissolving microneedles were used to deliver ketoprofen through rat skin [112]. The pre-treatment of pig skin with solid silicone microneedles resulted in the prolonged enhancement of the transdermal permeation of ketoprofen compared to untreated skin [113]. Similarly, the pre-treatment of mouse skin with microneedles (Derma Roller^®^) resulted in the improved permeation of celecoxib and doxorubicin from liposomal gels compared to non-microneedle-treated skin [114]. Dissolving microneedles loaded with an ME of celecoxib and alpha-linolenic acid were able to enhance transdermal flux and permeation through rat skin when compared to solutions of free celecoxib and the celecoxib ME alone. In addition, anti-inflammatory activity was found to be significantly increased following treatment with microneedle systems compared to systems without microneedles [115]. Microneedle arrays have also been used to enhance the transdermal permeation of other NSAIDs, such as diclofenac [116,117], ibuprofen [118,119], and acetaminophen [116].

### 2.4. Solubility Enhancement

#### 2.4.1. Supersaturation and Co-Solvency

Supersaturated drug delivery systems are formulations containing a drug in a concentration higher than its solubility, resulting in increased thermodynamic activity and enhanced drug movement from the vehicle into the skin [120,121]. Solution saturation can be increased by either decreasing the drug solubility or increasing the drug concentration in the formulation, which can be achieved through the evaporation of volatile solvents, the binary mixing of solvents (co-solvency), heat processing, and chemical processes [122].

Supersaturation through evaporation at the skin’s surface following the application of a topical formulation containing alcohols is a common way of enhancing the formulation’s thermodynamic activity. However, due to the difficulty in controlling the evaporation process, the degree of supersaturation, and thus, the concentration of the drug in the formulation can be unpredictable. This can result in inconsistencies in the amount and rate of drug delivered [122].

The co-solvency method has been used with many different solvent combinations to enhance the solubility of poorly water-soluble drugs. Typically, a saturated solution is first created in the solvent the drug is most soluble in, and then, it is diluted with the other solvent where the drug is less soluble (typically water), which decreases the overall solubility of the drug in the solution, thus creating a supersaturated solution. The ratio of drug solubility in a specific co-solvent (*S_C_*) to its solubility in water (*S_W_*) is known as the co-solvency power (σ) and can be calculated using the following equation [123]:$$\sigma =\log\left(S_{C}/S_{W}\right)$$

The co-solvency approach is one of the most commonly used methods to enhance solubility and has been used to enhance the solubility of various NSAIDs, such as rofecoxib, valdecoxib, meloxicam, acetaminophen, and celecoxib [124,125,126,127,128,129]. The primary drawback to supersaturated drug delivery systems is their thermodynamic instability, which makes these systems liable to crystal formation and poses significant challenges if developing formulations for commercial applications [122].

#### 2.4.2. Formation of Inclusion Complexes (Cyclodextrins)

Cyclodextrin (CD) is a cyclic oligosaccharide of alpha-D-glucopyranose units, with the most common CDs, alpha-, beta-, and gamma-, containing 6, 7, and 8 units, respectively [130]. CDs are cone-shaped molecules that have a hydrophilic outer surface and a lipophilic central core. Inclusion complex formation occurs when lipophilic drugs enter the central pore and bind to the cyclodextrin molecule [131,132]. The hydrophilic nature of the outer surface of the CD molecule increases the aqueous solubility of the lipophilic drug that is encased inside, allowing it to be incorporated into various aqueous vehicles. Beta-CD is the most widely researched CD in the pharmaceutical industry, particularly for drugs such as NSAIDs, where the central pore diameter is ideal to accommodate the aromatic ring structure [133]. While B-CD has the lowest aqueous solubility of the three most common CD types, this can be improved through modification of the molecule via substitution of the hydroxyl groups, with hydroxypropyl derivatives being commonly used [134]. Beta-CDs have been successfully used to increase the solubility of other coxib NSAIDs, such as celecoxib [131,135,136], rofecoxib [137], and valdecoxib [126], as well as many traditional NSAIDs (ketoprofen [138], tolfenamic acid [139], meloxicam [140], and piroxicam [141]). However, while CDs can improve the solubility of NSAIDs in a formulation, the transdermal permeation of CDs when used alone is limited [142], necessitating the addition of other permeation-enhancing excipients when aiming to achieve systemic drug concentrations.

#### 2.4.3. Eutectic Mixtures

A eutectic mixture is formed by co-melting a combination of two or more components that are highly miscible in their liquid state, but have limited miscibility in their solid state, to form a new compound where the melting point is lower than either of the individual compounds [143]. Solubility can be increased using eutectic mixtures by combining a drug with a hydrophilic carrier compound, or through combination with a more soluble drug [144]. NSAIDs, such as meloxicam, aceclofenac, and flurbiprofen, have been combined with caffeine in eutectic mixtures to improve the dissolution rate for oral delivery [145]. Deep eutectic solvents (DES) form liquids when combined at the correct ratio due to the lower melting point of the eutectic mixture, and have been gaining popularity due to their low costs and being environmentally friendly [146]. DES hydrogels containing ibuprofen were able to improve transdermal permeation across human skin compared to a conventional ibuprofen hydrogel [147]. Similarly, meloxicam delivery through guinea pig skin was enhanced using a eutectic mixture with thymol [70].

#### 2.4.4. Solid Dispersions

A solid dispersion is a formulation where the drug is dispersed in a solid carrier matrix. The choice of dispersion matrix depends on the properties of the drug and the desired characteristics of the formulation, with a wide variety of different materials being utilised to form solid dispersions, with common examples including cellulose HPMC, synthetic polymers, polyethylene glycol (PEG), Tween^®^ 80, poloxamer, and urea [148]. The incorporation of a hydrophilic matrix increases the surface area and contributes to the enhanced solubility of the drug in aqueous vehicles [149,150]. In addition, solid dispersions may help to reduce the aggregation of drug particles in various formulations. Solid dispersions were demonstrated to enhance the transdermal permeation of celecoxib through rabbit skin compared to conventional creams [151]. The transdermal permeation of rofecoxib through rat skin was also enhanced using solid dispersions of rofecoxib-PEG 4000 compared to standard rofecoxib in MEG formulations [152].

#### 2.4.5. Polymorphic Forms

Drugs often exist in various solid-state forms, which can be classified as polymorphic, pseudopolymorphic, amorphous, polyamorphous, and pseudopolyamorphous. Polymorphic forms have the same chemical composition but vary in molecular structure (crystalline packing of molecules or conformation) [153,154]. Pseudopolymorphs contain additions from solvents (solvates), or water (hydrates) incorporated into their molecular structure. Amorphous forms lack any distinct structure or order. The degree of crystallinity affects the surface energy, with less crystalline forms having higher surface energy. Consequently, these forms become more reactive and have better solubility, but are less stable. Since polymorphs vary in crystalline structure or conformation, it is widely accepted that the different polymorphs of a drug may result in different physicochemical, thermodynamic, and structural properties, including solubility, melting points, stability, and bioavailability [153]. It is therefore important when manufacturing drugs and developing drug delivery systems that we have a thorough understanding of which polymorphic forms exist, what their characteristics are, and which forms are in the drug delivery system. There are various methods for identifying and manufacturing the different polymorphic forms of a drug. Firocoxib has two known polymorphic forms, A and B. A report from the European Medicines Agency [155] stated that in the manufacturing process, only the stable monotropic form B is produced; however, no published papers are available regarding the various forms or the studies conducted. Recently, methods using supercritical carbon dioxide (scCO_2_) have gained interest as an inexpensive, easy to use, ecologically friendly alternative to traditional solvents (see below).

#### 2.4.6. Supercritical Carbon Dioxide (scCO_2_)

A supercritical fluid is a substance that is above its critical pressure and temperature. At the critical point, equilibrium is reached between the vapour and liquid states, and above the critical point, a fluid exists (supercritical fluid) that possesses both liquid- and gas-like properties [156]. Carbon dioxide is popular as a supercritical fluid due to its near-ambient critical temperature (31.1 °C) and pressure (73.8 bar) [157]. Carbon dioxide is also easily accessible, relatively inexpensive, non-toxic and non-flammable, recyclable, and environmentally friendly [158]. scCO_2_ has been used for many applications in the pharmaceutical industry, including extractions, nano- and microparticle production, and polymer processing and as a solvent [159]. The solubility of the drug in supercritical fluid can be altered and improved by varying the pressure or temperature of the system. Once solubilised in the scCO_2_, the drug can be recrystallised using rapid expansion (rapid expansion of supercritical solution; RESS) into its various polymorphic forms depending on the pressure and temperature. In addition, the recrystallisation process often yields smaller drug particles, which can increase drug solubility due to the increase in surface area, allowing greater interaction with the solvent [149,160]. One study used an RESS technique in scCO_2_ on carbamazepine and reported significant particle size reduction and the presence of four different polymorphic forms which could be selectively created through temperature and pressure selection [161]. Similarly, ibuprofen [162,163] and naproxen [164] have been explored using RESS techniques, although the products of these studies have not been evaluated in transdermal systems.

Supercritical CO_2_, despite having many advantages, is limited by the solubility of the drug in the supercritical fluid, and highly specialised equipment is generally required to increase the batch size for formulation production, which has been extensively outlined elsewhere [160,165,166].

#### 2.4.7. Prodrugs

A prodrug is a pharmacologically inactive derivative of an active drug that is designed to possess more desirable characteristics for transdermal permeation through the skin. Once the prodrug has entered the skin, it will undergo metabolism, where the active form of the drug will be enzymatically cleaved, and will be able to elicit its action in the body [167]. Prodrugs of NSAIDs are useful to reduce adverse effects, particularly related to oral dosing, or to increase the solubility of poorly soluble forms (i.e., coxib NSAIDs) [168,169]. There are a few formulations consisting of NSAID prodrugs commercially available for human use, including the COX-2-selective inhibitor prodrug of valdecoxib (parecoxib) [170,171]. A prodrug of celecoxib (celecoxib-butyryl) was shown to have greater relative bioavailability than standard celecoxib when taken orally [172]. Similarly, nano-sized celecoxib prodrugs were shown to have in vitro anti-inflammatory activity by inhibiting TNF-α synthesis in a lipopolysaccharide-activated cell line [173]. However, these formulations were found to be less effective at decreasing swelling than standard celecoxib using a mouse collagen-induced arthritis model due to slow release from the prodrug. The pharmacokinetics of a paracetamol prodrug, propacetamol, have also been described in dogs, although the efficacy of this product may be less effective in dogs than in humans [174].

## 3. Vesicles and Nanotechnology

### 3.1. Liposomes

Liposomes are vesicles made of lipid bilayers composed of mainly phospholipids surrounding a central aqueous phase [175]. Cholesterol is frequently included to improve phospholipid packing, regulate membrane fluidity, modulate drug release, and enhance the stability of the liposome [176]. Lipids can be structured as either in a single bilayer (unilaminar) or in multiple bilayers (multilaminar). The phospholipids have polar head groups and non-polar tails, which allow the incorporation of both hydrophilic and lipophilic molecules in either the central core or amongst the lipid tails, respectively. There are several potential outcomes when a liposome is applied to the skin. The drug may be released from the liposome on the skin surface and then diffuse through the SC following the normal intercellular diffusion pathway. The liposome may be absorbed into the skin, where it interacts and disrupts the intercellular lipid lamellae, acting as a permeation-enhancing agent to increase drug permeation through the SC. Generally, liposomes cannot traverse the SC layer intact as they are too rigid, although it has been demonstrated that ultra-deformable liposomes (transfersomes, see below) are able to move through the SC intact to the underlying dermis [177,178]. Finally, transappendageal transport can be used to bypass the SC layer through the shunting of liposomes through hair follicles and glands [177]. The ability to permeate the skin is influenced by the type of lipid, the size of the vesicle, and the integrity of the skin. The type of lipid in the vesicular membrane can affect many aspects of the liposome’s performance, such as vesicle size, stability, and skin permeation [179], with phospholipid liposomes of lower cholesterol content being preferable for dermal applications [177,180]. Liposomes come in a range of sizes, with multilaminar vesicles typically being larger than unilaminar vesicles. Smaller liposomes (≤120 nm) are better able to penetrate deeper skin layers, while 70 nm liposomes were found to be optimal for TD [181].

Liposomal delivery systems have been used to enhance the transdermal permeation of various NSAIDs. For example, the TD of ketoprofen permeation was enhanced using a 1% diclofenac liposomal gel compared to both 1% and 2% diclofenac emulsion gels [182], and the flux of celecoxib through rat skin was significantly higher in a liposomal formulation compared to a saturated solution [183].

### 3.2. Transfersomes

Transfersomes are ultra-deformable liposomes that are composed of phospholipids and a membrane-softening edge activator, which are usually surfactants, such as Tweens, Spans, and bile salts, in an aqueous suspension [184]. The increased flexibility of the transfersome membrane allows the vesicle to deform and pass through narrow pores and junctions in the SC, passing through to the deeper dermis [185]. The osmotic gradient between the water-rich epidermis and the transfersome may help facilitate its transdermal permeation [185]. Transdermal transfersome formulations have been used to enhance the permeation of meloxicam compared to conventional liposome formulations and suspensions [186]. Ketoprofen from transfersome gels provided effective relief from pain in knee osteoarthritis [187]. Transfersome gels containing celecoxib showed significantly higher drug permeation through human skin compared to aqueous suspensions [188], and transfersome hydrogels of lornoxicam increased drug flux and anti-inflammatory activity compared with standard lornoxicam hydrogel and indomethacin gels [189].

### 3.3. Ethosomes

Ethosomes are another lipid-based vesicular system, comprising high levels of ethanol (up to 45%) [190]. Binary ethosomes can be created through the inclusion or another type of alcohol, typically PG or isopropanol, which increases the stability and dermal drug delivery of ethosomes [191]. A third type of ethosome, which is a combination of a transfersome and an ethosome called a transethosome, contains both ethanol and an edge activator [192]. Similar to the transfersome, the ethosome is deformable, permitting deeper penetration into deeper skin layers. Compared with traditional liposomes, ethosomes have a greater drug encapsulation and are smaller in size [193], as was demonstrated with etodolac ethosomes [194]. The transdermal permeation of celecoxib through excised human skin was enhanced 9-fold using ethosomes compared to a celecoxib suspension [188]. Naproxen in ethosomal gels made from phosphatidylcholine, cholesterol, ethanol, and a Carbopol^®^ polymer showed increased transdermal permeation over a 24 h period [195].

### 3.4. Niosomes

An alternative vesicular system to enhance transdermal drug delivery is the niosome, which is a vesicle containing a non-ionic surfactant (Figure 2). Because non-ionic surfactants are composed of a polar head group and hydrophobic tail, niosomes can incorporate both hydrophilic and hydrophobic molecules [48]. While many different surfactants may be used in niosomes, polysorbates (Tweens), sorbitans (Spans), and polyoxyethylene glycol alkyl ethers (Brij^®^) are most commonly used [196]. Niosomal gels exhibited excellent entrapment efficiency and increased transdermal permeation of lornoxicam, an NSAID, through rat skin, compared to free drug solutions, and provided a significant enhancement in anti-inflammatory action in a rat paw oedema model [197].

### 3.5. Nanoparticles

The term nanoparticle (NP) encompasses a wide range of nano-sized (10−1000 nm) drug delivery systems made up of natural or synthetic polymers, gels, or lipids [198]. Because the small size of the particles gives a large surface area, often, these systems have high drug solubility and improved bioavailability [198,199]. Polymeric NPs are created using biocompatible polymers, such as polystyrene, gelatin, and polylactic acid, with the drug dissolved into or encapsulated inside [200]. Surface coatings on the particles can be used to modify how the particle interacts with the cells in the body and can influence the transdermal permeation and bioavailability of the nanoparticle [201]. Nanoparticles of chitosan and egg albumin were shown to prolong the duration of drug release of aceclofenac from a transdermal gel formulation and significantly enhanced the anti-inflammatory action in a rat paw oedema model [202]. A similar effect was demonstrated with nanoparticles containing carprofen, where inflammation was significantly decreased in a mouse ear oedema model, compared with carprofen solutions [203].

A solid lipid nanoparticle (SLN) is a dispersion of solid lipids in an aqueous phase stabilised by surfactants and emulsifiers [175]. SLNs are small in size (<100 nm) [204] and have a near-perfect crystalline structure of lipids [205]. They have been used to enhance the permeation of ibuprofen through rat skin [206]. However, drug loading in an SLN is often lower than in other lipid-based systems because of the highly crystalline structure [205], which may lead to crystallisation during storage and the precipitation of previously dissolved drug [207]. Gels containing SLNs of cetostearyl alcohol, polysorbate 80, and sodium hydroxide have been used to enhance the permeation of ibuprofen through rat skin compared to the commercially available Nurofen^®^ 5% gel [206]. SLNs have also been used to enhance celecoxib permeation through rat skin and significantly increase anti-inflammatory activity in a rat paw oedema model compared to a standard celecoxib gel [208].

### 3.6. Nano- and Microemulsions

A nanoemulsion (NE) and a microemulsion (ME) are isotropic dispersions of two immiscible liquids, oil and water, stabilised using a surfactant, that form a thin layer over the dispersed droplet which is suspended in the continuous phase. An additional co-surfactant is often required to reduce and maintain the low interfacial tension of the droplets, allowing an incredibly small droplet size to be achieved. NEs and MEs may be classed as oil-in-water (O/W), water-in-oil (W/O), or bicontinuous/multiple, depending on which phase is dispersed (Figure 3) [209]. Typically, O/W-type emulsions are the most common in TD systems, as they offer the ability to deliver lipophilic drugs through the skin [210]. Both NEs and MEs are small (<200 nm), resulting in transparent systems due to the particle size being smaller than the visible-light wavelength [211]. Because of the small droplet size, both emulsion types are visually clear liquids and virtually indistinguishable to the eye; however, they display very different thermodynamic properties. When at the correct concentration and temperature, MEs form spontaneously and are thermodynamically stable systems, whereas NEs require high energy input to form and are thermodynamically unstable systems, making long-term formulation stability a greater challenge than with MEs [212].

NEs and MEs have been shown to enhance the transdermal permeation of poorly water-soluble drugs through a number of proposed mechanisms. The small droplet size increases the surface area for drug solubilisation and the thermodynamic activity towards the skin, and the components of the emulsions, such as the surfactants and co-surfactants, may act as permeation-enhancing agents by altering the barrier function of the SC [42,213]. In addition, high-water-content NEs and MEs may maintain or increase skin hydration, further enhancing SC permeation. There have been extensive studies using both NEs and MEs to enhance the transdermal permeation of various NSAIDs, including many coxib NSAIDs. The transdermal absorption and bioavailability of celecoxib in rats were significantly increased compared to oral capsules using both liquid NE- and NE-based gel systems [214]. Similarly, the transdermal permeation from an MEG increased the transdermal flux and anti-inflammatory effect compared to celecoxib cream in rabbits and rats, respectively [215].

A similar system, termed the self-microemulsifying drug delivery system (SMEDDS), comprises oil, surfactant, and co-surfactant phases, which are loaded with the drug, and when combined with a biological aqueous phase (e.g., stomach fluid, mucosal or skin secretions), spontaneously form an ME [216]. While this route has been highly advantageous for delivering highly lipophilic drugs orally, there is limited research on its application for TD. However, the potential for SMEDDS to be used in a TD application was demonstrated using indomethacin on ex vivo rabbit skin, where permeation was enhanced compared to that using a conventional ME [217].

## 4. MEs for Transdermal Drug Delivery

There are several key advantages to using MEs as TD systems, including solubility enhancement for poorly water-soluble drugs and increased transdermal permeability due to their small droplet size. They also have the ability to incorporate both hydrophilic and lipophilic drugs into their various components. The ME will spontaneously form when the components are combined in suitable proportions, negating the need for expensive high energy inputs and simplifying the manufacturing process. These factors (enhanced transdermal permeation, increased drug solubility, good thermodynamic stability, and ease of manufacture) make transdermal MEs popular systems, particularly for difficult-to-formulate, poorly water-soluble drugs, such as NSAIDs.

### 4.1. Components of an ME

MEs are four-component systems, containing an oil and an aqueous phase, the latter of which is usually water, as well as a surfactant and a co-surfactant. The components of the ME system, particularly the oil and surfactant phases, are often selected on the basis of the solubility of the drug of interest. A wide range of oils have been explored in transdermal MEs, including fatty acids and fatty acid esters, such as OA and IPM, alcohol esters, medium-chain triglycerides, and many different essential and plant oils [218,219,220,221,222,223,224].

Surfactant selection is very important for the formation and stability of an emulsion. Surfactants, as discussed previously, consist of a hydrophilic polar head group with a lipophilic tail. The amphiphilic nature of the surfactant allows it to form a monolayer around the droplets, decreasing the interfacial tension, providing stability, and preventing phase separation. Commonly selected surfactants in MEs include polysorbates (Tween^®^ 80, Tween^®^ 20), sorbitan esters (Span^®^ 20), and Plurols (Plurol^®^ Oleique, and Plurol^®^ Isostearique), although many other surfactants have been used [62,225]. The co-surfactant chosen for the ME system is designed to help stabilise the interfacial membrane of the droplets and further decrease surface tension. Short-chain alcohols, such as ethanol and isopropanol, are commonly used, with the length of the alcohol chain affecting the size of the ME region [219]. PG, DGME, and PEGs have also been extensively used as co-surfactants in various ME systems [62,226]. The final component of the ME is the aqueous phase, which is commonly water. Preservatives, pH modifiers, water-soluble permeation enhancers, or viscosity-enhancing agents may be added to the aqueous phase to achieve the desired formulation characteristics or improve stability [226].

### 4.2. Types of MEs

The type of ME (O/W, W/O, or bicontinuous) is influenced by the type and relative amounts of each of the four components in the system, but most importantly, the surfactant and co-surfactants. It should be noted that the combination of water, oil, and the surfactant may also lead to the formation of other phases, including coarse emulsions and liquid crystals (lamellar, cuboidal, or hexagonal), although these phases are often visually distinguishable from MEs due to being opaque (coarse emulsions) or more viscous (liquid crystal phases) [209].

Several methods have been proposed to assist in selecting the most appropriate surfactants for emulsion systems. One of the simplest rules to guide surfactant selection is Barcroft’s rule, which states the following:
‘*The phase in which the emulsifier is more soluble tends to be the dispersion medium [continuous phase]*’. [227]

Following this rule, a water-soluble surfactant should form an O/W emulsion, and an oil-soluble surfactant should form a W/O emulsion [228]. However, while useful as a guide, it should be noted that this rule has its limitations and may be incorrect, particularly at low surfactant concentrations [228]. The hydrophilic–lipophilic balance (HLB) concept is a semi-empirical scale designed for selecting surfactants in emulsion systems. This scale assigns a number (HLB value) based on the proportion of hydrophilic to lipophilic groups in the surfactant molecule [229]. A hydrophobic surfactant will have an HLB value below 8, and a hydrophilic surfactant will have an HLB value greater than 8. Ionised surfactants also have higher HLB values due to their charged head group having a strong interaction with water molecules [230]. In general, W/O emulsions require an HLB value of <6, and O/W emulsions >8, although the exact HLB value needed for a system depends on the nature of the oil phase used. Mixtures of surfactants with different HLB values can be used to achieve the optimal HLB value for a given system [231], although as other parameters, such as temperature and the presence of electrolytes, can affect the emulsification process, this method should be used as a guide only [232]. Alternative theories, such as hydrophilic–lipophilic deviation (HLD), that incorporate formulation salinity and temperature may also be used to inform surfactant selection and the optimal conditions for ME formation [233,234,235]. Additionally, the packing parameter (*PP*) theory is often used to estimate the curvature of the interfacial film based on the geometry of the surfactant molecules and packing at the oil–water interface, and can be explained by the following equation [236].
$$PP=\frac{V}{\alpha l}$$
where *V* is the volume of the surfactant tail, *α* is the area of the surfactant molecule head group, and *l* is the length of the surfactant tail.

Following this theory, when *PP* < 1, a positive curvature is created and O/W MEs are created, with the opposite being true for *PP* > 1. When *PP* = 1, a neutral curvature is produced, giving rise to bicontinuous or liquid crystalline structures [236]. However, this overly simplified system is also limited in predicting the microstructure in these systems with complex interactions and with changes in temperature.

### 4.3. Preparation of MEs

#### 4.3.1. Phase Titration Method

MEs are spontaneously forming systems when the four components are combined at the appropriate ratios. The aqueous phase titration is a common method to spontaneously form MEs, where the oil and surfactant phases are first combined, followed by slow titration with the aqueous phase until a transparent ME is formed [237]. Pseudoternary phase diagrams are often used to study the phase behaviour of ME systems and are used to define the boundaries between different phases, allowing the determination of the composition at which MEs form (termed the ME region) (Figure 4) [224]. In a pseudoternary phase study, the surfactant and co-surfactant are considered a single component, called the surfactant mix (Smix), and are combined at a pre-determined fixed ratio. This type of study is performed by combining Smix and oil together at ratios ranging from 1:9 through to 9:1 (Smix/oil), and then, the aqueous phase is slowly titrated, often using moderate agitation or stirring, until a phase change occurs. Phase changes can be easily ascertained by the turbidity and viscosity of the formulation, with MEs being transparent and coarse emulsions being turbid [216,238]. When a phase change occurs, the amount of aqueous phase added to reach that transition can be calculated, and the ratio of all components can be plotted on a phase diagram [239]. Once the boundaries for the ME regions have been established, the compositions where MEs form can be easily determined [237]. The selection of a point within the ME region/s can be used as the composition for an ME formulation [240].

#### 4.3.2. Phase Inversion Method

Phase inversion methods work through changing the spontaneous curvature of the ME droplets through changes in temperature (phase inversion temperature (PIT)) (Figure 5) or changes in ME composition (phase inversion composition (PIC)) [241]. The transition of the surfactant film curvature from positive to negative with an increase in temperature will result in the formation of a W/O ME due to the dehydration of non-ionic surfactant molecules [242]. Similarly, decreasing temperatures will favour the formation of O/W MEs. The PIT method can only be used with non-ionic surfactants, although phase inversion, using changes in composition, pH or salts, may be used with MEs containing other types of surfactants [243].

### 4.4. ME Gels (MEGs)

One of the main drawbacks of MEs for transdermal application is their low viscosity, so gelling agents, such as polymers and gums, are often included in the final formulation to aid with adhesion to the skin and improve the formulation feel and ease of application [244]. Many different gelling systems have been used with transdermal MEs, including hydroxy propyl methyl cellulose (HPMC) [245,246], Xanthan gums [247,248], and Carbopol^®^ polymers [30,69].

The transdermal delivery of NSAIDs using ME and ME gel-based delivery systems has been explored extensively in the literature using a range of different NSAIDs, including a number of coxib NSAIDs. It has been demonstrated that celecoxib in an MEG containing triacetin oil, Tween^®^ 80, and Transcutol^®^ (DGME), gelled with a Carbopol^®^ 934 polymer, could improve transdermal permeation compared to a conventional gel, as well as increase bioavailability compared to a commercially available oral tablet [30]. The anti-inflammatory activity of celecoxib was increased with MEG systems of triacetin, cremophor RH 40, PEG 400, and Carbopol^®^ polymers compared to a conventional gel [9]. The importance of ME type on transdermal permeation was demonstrated when O/W MEs enhanced the permeation of meloxicam through human cadaver skin, whereas W/O MEs did not [249].

Transdermal MEG formulations containing rofecoxib had faster drug release during in vitro studies, while MEGs containing rofecoxib solid dispersions had greater cumulative drug permeation through excised rat skin [152]. In addition, MEGs containing a rofecoxib solid dispersion had significantly enhanced anti-inflammatory activity compared to that of a conventional gel formulation [152]. Similarly, the anti-inflammatory effect of ketoprofen was enhanced using MEG formulations compared to conventional gels at 4 h following the induction of oedema [250]. The controlled drug delivery of NSAIDs has also been investigated using MEs and MEGs. It was reported that MEs of Capryol^®^, Labrasol^®^, Transcutol^®^, and water could enhance diclofenac’s skin permeation and sustain its anti-inflammatory activity in a rat paw oedema model long after formulation removal due to the formation of a skin drug depot from the ME system [251]. Transdermal MEs incorporated into patch systems were also able to provide the controlled release of dexibuprofen through rat skin [252].

## 5. Conclusions

The TD of NSAIDs has many benefits, including that it reduces adverse effects; avoids hepatic first-pass metabolism; is easy to administer; and is non-invasive. However, the solubility limitations of NSAIDs, particularly the selective coxib class, make incorporating the drug into a suitable vehicle challenging, and limit their permeability through the skin. This review provides a summary of the various strategies used to enhance the permeation of NSAIDs through the skin and increase their overall solubility in TD vehicles. Advanced nano-sized TD systems, such as liposomes, niosomes, transfersomes, ethosomes, solid lipid nanoparticles, and nanoparticles, were reviewed, with particular emphasis on the ME system. MEs have many advantages for transdermal delivery and have been shown to be effective in delivering a wide range of poorly water-soluble drugs through the skin. Because of their small droplet size (<150 nm) and large surface area, the ME system offers both increased drug-solubilising capacity and transdermal permeation. In addition, systems of oil, surfactant, co-surfactants, and water are spontaneously formed when their components are present in the right quantities, making their manufacture simple, cost-effective, and easy. ME and MEG systems have been extensively studied in the literature; however, it is important that each new ME and drug combination be evaluated and optimised in the species of interest, as it is known that permeation data between species cannot be extrapolated, and drugs may behave differently in a vehicle depending on the physicochemical properties of the drug itself. Furthermore, while many NSAIDs have been evaluated in ME systems, limited studies have been conducted assessing their clinical efficacy and safety in target species, although these systems show substantial promise and value as effective transdermal systems for poorly soluble NSAIDs.

## Figures and Tables

**Figure 2 pharmaceutics-16-00675-f002:**
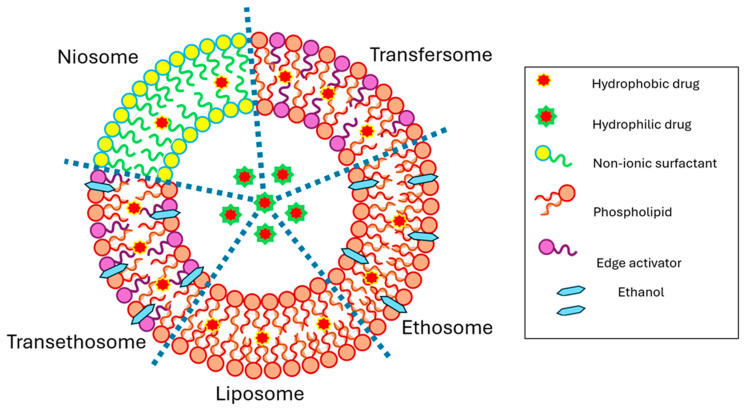
Structure of various nano-sized vesicle systems. Liposomes are composed of phospholipids. Ethosomes are composed of phospholipids and ethanol. Transfersomes are composed of phospholipids and edge activators. Transethosomes are composed of phospholipids, edge activators, and ethanol. Niosomes are composed of non-ionic surfactants.

**Figure 3 pharmaceutics-16-00675-f003:**
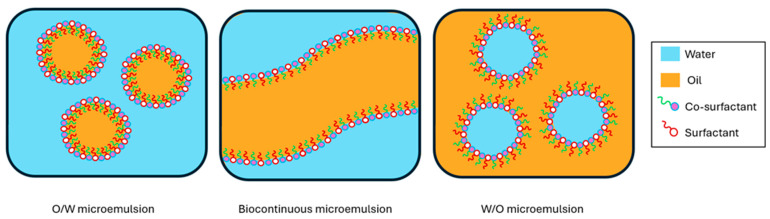
Schematic representation of the possible ME types. O/W is oil-in-water, W/O is water-in-oil.

**Figure 4 pharmaceutics-16-00675-f004:**
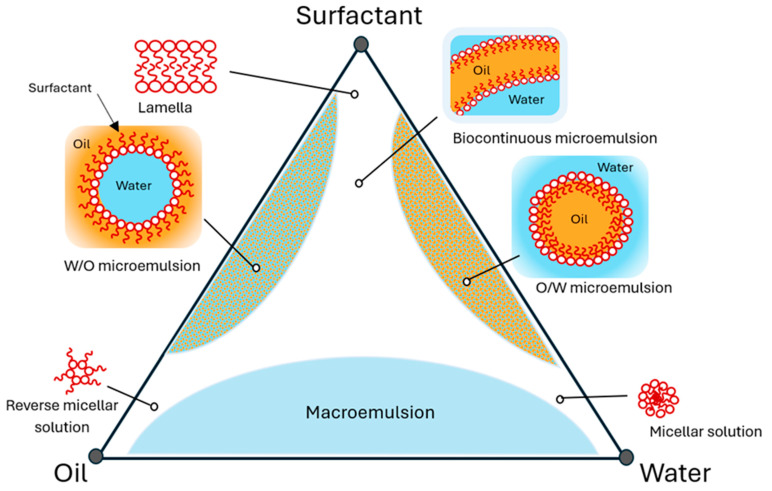
Theoretical pseudoternary phase diagram of an oil, surfactant (surfactant and co-surfactant as a combined mixture), and water system depicting the possible structures and phases that may exist.

**Figure 5 pharmaceutics-16-00675-f005:**
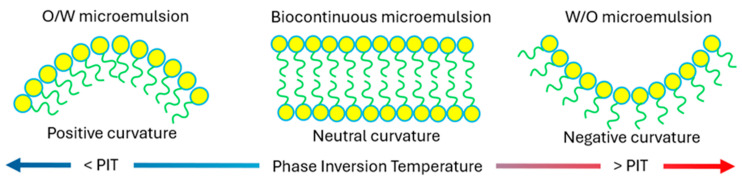
Representation of the microstructural changes and changes in the curvature of the ME interfacial layer when the temperature is increased or decreased from the phase inversion temperature.

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
