# Peer review of "Strategies to Improve the Transdermal Delivery of Poorly Water-Soluble Non-Steroidal Anti-Inflammatory Drugs"

_pharmaceutics, 2024, doi:10.3390/pharmaceutics16050675_

Round 1
Reviewer 1 Report (Previous Reviewer 2)
Comments and Suggestions for Authors
I am satisfied with this revised version. Thank you for sparing time and correcting point by point.
Author Response
Thank you – your feedback was appreciated.

Reviewer 2 Report (Previous Reviewer 3)
Comments and Suggestions for Authors
Please see file attached.

1. It seems that the authors took time to improve the paper substantially. However, there are still some concerns that need to be dealt with. The English used is not of a high quality.
2. The English used by the authors still vary between US and UK English. Specific examples have been given previously as well as below.
3. There are still numerous abbreviations used. The authors are furthermore not consistent with their abbreviations. It is clear that they have not aligned the different sections written by different authors. Some use for example, CPE or CPEs; whereas others use PE or PEs, which were NOT defined.
4. The authors still have not defined various abbreviations and they interchange between abbreviations and not abbreviating the words. There is no consistency.
Author Response
All the issues raised by the reviewer were addressed in a point-by-point manner and are outlined in the attached document.

Round 2
Reviewer 2 Report (Previous Reviewer 3)
Comments and Suggestions for Authors
NA
This manuscript is a resubmission of an earlier submission. The following is a list of the peer review reports and author responses from that submission.
Round 1
Reviewer 1 Report
Comments and Suggestions for Authors
The authors are describing different strategies to improve the transdermal delivery of NSAIDs. I am not sure if this review article is significantly contributing to the field since another review with a similar topic was published in 2018 (Advanced Drug Delivery Systems for Transdermal Delivery of Non-Steroidal Anti-Inflammatory Drugs: A Review - PubMed (nih.gov)), but a broader scope of enhancement strategies is indeed described through the text.
My comments and suggestions for the manuscript are the following:
Major comments:
1) The title prepares the reader for a text connected with the solubility of NSAIDs (there is the phrase: of poorly water soluble NSAIDs). I would suggest either further discussing the issue of solubility in the manuscript and categorizing NSAIDs according to their solubility or just completely omitting the phrase poorly water soluble from the title.
2) The review may be interesting for readers working with transdermal delivery strategies and any reference to the skin lipids should be accurate. I'm afraid I have to disagree with the use of expressions like lipid bilayers of stratum corneum (line 67). The lipids of the stratum corneum are very well characterized through the years and it has been shown that their special characteristics come from their multilayer arrangement. I would highly recommend revising the part of the manuscript describing the stratum corneum lipids to be scientifically correct.
3) Many chemical penetration enhancers are described, and others are missing (e.g. alkyl and benzoic acid esters, Propyleneglycolmonolaurate (PGML), etc). Similarly, various other types of physical penetration enhancers (e.g. Ultrasound, Laser radiation, and photomechanical waves) are not mentioned in the text. Were there any specific reasons to exclude those enhancers or enhancing strategies or they haven’t been used for the transdermal delivery of NSAIDs?
4) The authors include molecules like hydrocortisone, doxorubicin, and carbamazepine in the text. Are these molecules NSAIDs? Please check all molecules if they belong to NSAIDs.
Minor comments:
1) Typos:
line 16: change NSIAD to NSAID
line 56: change tropical to topical
line 65: change with to when
line 95: change formulas to formulation
line 106: NSAIDS to NSAIDs
2) Inconsistency with abbreviations:
lines 47 and 49: TD delivery systems. Please delete the word delivery.
lines 108 and 112: Please use PG instead of Propylene glycol (Propylene glycol is already abbreviated as PG in line 79).
lines 120 and 123. Please use the registered trademark symbol ® after the word Transcutol.
line 131: Please use OA instead of Oleic acid (Oleic acid is used in line 115, so it is better to define abbreviation in line 115 and then just use abbreviated form).
line 134: Does PE stand for Penetration Enhancer or is it CPE? Please define it appropriately.
line 502: Please change Microemulsions to ME.
line 507: Please change Stratum Corneum to SC.
3) Inconsistency with references:
line 70: ref Haque and Talukder, 2018. Please use normal reference numbering [24].
lines 91, 375, 378, 411, 414, 440, and 647: Please use the First Author et al format.
lines 493 and 588: Please correct the reference error.
4) Formatting inconsistencies:
The authors have categorized paragraph 2. Enhancement of transdermal drug delivery (line 51) into 2.1. Chemical penetration enhancers, 2.2. Physical penetration enhancers, 2.3. Bypassing the stratum corneum and 2.4. Solubility enhancement. In segment 2.1. the authors are giving a brief introduction about the CPE and then in a new subsection 2.1.1. Categories of CPE are described. However, this formatting pattern is not followed in paragraphs 2.2 and 2.4., where a direct classification with sub-headings is noticed. In addition, 2.3. lacks any categorization. I would recommend using a similar formatting style to all paragraphs.
Reviewer 2 Report
Comments and Suggestions for Authors
The review work “Strategies to improve the transdermal delivery of poorly water soluble NSAID” focuses on the various strategies employed with poorly-water soluble NSAIDs to improve their solubility in TD delivery systems and permeation through the skin. There are reviews on generalized strategies, however here authors tried to be specific for NSAIDs, which makes it unique. The review discuss only limited data over here, indepth literature coverage is missing and even flow need to reevaluate. The whole review need to rewrite with careful consideration of revised TOC and discussing critical parameters with findings.
The constructive comments are:
1. There is need of revision in introduction and addition of illustrations of techniques/strategies which is covered in this review.
2. Please add preface at all headings, 2, 3 etc and then start subheading.
3. Need to compile more recent data which prove this strategies are superiors.
4. Please discuss limitation of each strategies, with two or three sentences.
5. What about the regulatory concerns for this strategy?
6. Line: 563-564. What is this about: ‘The phase in which the emulsifier is more soluble tends to be the dispersion medium 563 [continuous phase].’ [201]
7. The method of preparation is not connected at all.
8. The addition of Patented technology, patented formulations technology, safety of material included, clinical trial if any, future perspectives must be there in review.
Reviewer 3 Report
Comments and Suggestions for Authors
Please see attached file.

The English language is readable, however, there are numerous spelling and grammatical errors that need to be addressed. The authors also vary between UK and US English, for example synergises vs. synergizes OR characterised vs characterized. They need to understand the difference between a colon (:) and a semicolon (;). There are furthermore abundant abbreviations that leave the reader a little confused sometimes; and one needs to page back several times to remind oneself what an abbreviation means.
Reviewer 4 Report
Comments and Suggestions for Authors
The review article titled "Strategies to Improve the Transdermal Delivery of Poorly Water Soluble NSAIDs" is, first and foremost, well-written and comprehensively supported by literary sources. The general strategies to enhance skin permeation of NSAIDs are adequately described. However, I have some concerns. Regarding the title, we would expect more emphasis on NSAIDs. Since strategies for improving transdermal delivery are detailed in many review articles, the authors' opinions about the potential of each approach would be beneficial. From this point of view, the conclusion is not adequately supported by the given information.
There are also some conceptual lacks. Surfactants are described generally, with well-known facts about composition and classification in two chapters: about penetration enhancers as well as in the section about microemulsions. This seems redundant.
Solid lipid nanoparticles are discussed in Chapter 3.5, while Chapter 3.6 is entitled "Nanoparticles," so we would expect it to contain solid lipid nanoparticles since they are one class of nanoparticles. It would be better to move them.
The same applies to nano- and microemulsions, which are described in subchapter 3.7, and then the entire Chapter 4 is dedicated to microemulsions. Additionally, the rationale why microemulsions are emphasized is missing, especially concerning NSAIDs.
An explanation of why microemulsions are thermodynamically stable and nanoemulsions are not should be provided.
There are also some typographical errors.
Comments on the Quality of English Languageline 56: tropical instead of topical
line 65:: with apply to the skin
line 95:formualtion instead of formulars
...
Reviewer 5 Report
Comments and Suggestions for Authors
The article presents an excellent review of transdermal drug delivery and NSAIDs, containing 219 references which involves a significant amount of work. Although not new, the theme is revised over the already known review articles. The article's main focus is on microemulsions used with coxibs and this should be highlighted.
Some reviews found on the subject are already five years old: doi:10.1080/03639045.2019.1680996; doi:10.2174/1567201815666180605114131
CPE are very well developed and several reference examples are included.
The search seems exhaustive and systematic but it should be explained the search methodology used. The subject is not new and this is a valuable information to include in the introduction to increase the credibility of the research.
Patents are not included in this review and are relevant (as in this article doi: 10.1080/13543776.2018.1519025 or the newest doi:10.3390/pharmaceutics15122762) to include and update.
The figures included are adequate.
Please consider including a summary table on the main topic resuming the NSAID and the enhancer mechanism among others.
Specific comments:
line 55: replace '...tropical formulation...' by topical formulation
line 493, line 588: correct the reference error
Contextualize and refer in the text figure 3 and figure 4
subchapter 2.1. Chemical penetration enhancers should refer the increase of partitioning of the drug into skin as a general penetration-enhancing mechanism
ref. 187 is not complete
Round 2
Reviewer 1 Report
Comments and Suggestions for Authors
The manuscript has been updated - more or less - with the suggeted corrections.
Minor comments:
a) Please revise the use of TD abbreviation. According to lines 30-31, TD means "trans-dermal drug delivery". In lines 291, 293, 598 and 759 the abbreviation is not appropriately used. For example in line 291 you are writing "TD drug delivery" which is translated like "trans-dermal drug delivery drug delivery". and it should be corrected.
b) The abbreviation NE is explained as Nanoemulsions and the abbreviation ME is explained as Microemulsions (line 564). The use of NEs and MEs is not needed later on (580, 605. 612, 615, 618, 625, 639, 686, 688 and 692).
c) The title for paragraph "4.3 Methods of preparation" is missing.
d) The formating inconsistencies suggested as comment 4 in my previous report were not changed. I leave the paragraph formatting to the journal proof-reading.
Reviewer 2 Report
Comments and Suggestions for Authors
Dear Authors,
The work need to updated with recent literature as suggested and in reply I have not found substantial revisions only reply is
"As noted above, the author guidelines state: ‘No new, unpublished data should be presented’:
Please under stand this line is for specially research work, where ethics are there, for review you need to revise and update the manuscript with the latest concert literature.
Please try revised as suggested.
Reviewer 3 Report
Comments and Suggestions for Authors
Please find attached

I indicated in the previous round that the English language is readable, however, there are numerous spelling and grammatical errors that need to be addressed. The authors also vary between UK and US English, for example synergises vs. synergizes OR characterised vs characterized. They need to understand the difference between a colon (:) and a semicolon (;). There are furthermore abundant abbreviations that leave the reader a little confused sometimes; and one needs to page back several times to remind oneself what an abbreviation means.
The authors replied: We have used UK spelling (the author guidelines did not specify). It is commonly accepted that an abbreviation is defined when first used, then used thereafter – we are unsure what the reviewer requires us to do regarding the abundant abbreviations.
The problem is not that the authors used UK spelling. The problem is that they vary between UK and USA spelling.
The second problem is not that they use abbreviations. The problem is that they use too many! This renders the paper difficult to read.
They also did not address the other problems noted.
Reviewer 4 Report
Comments and Suggestions for Authors
the authors duly considered my comments
